# Trend, burden and determinants of undiagnosed hypertension in the Horn of Africa: A systematic review and meta-analysis

**Muluken Chanie Agimas**[1]*, **Nebiyu Mekonnen Derseh**[1], **Fantu Mamo**[1], **Moges Tadesse Abebe**[2], **Tilahun Yemanu**[1], **Meron Asmamaw**[1]

**1** Department of Epidemiology and Biostatistics, Institute of Public Health, College of Medicine and Health Science, University of Gondar, Gondar, Ethiopia, **2** Department of Nursing, College of Health Science, Debark University, Debark, Ethiopia

* mulukensrc12@gmail.com

**Data Availability Statement:** All relevant data are in the paper and its Supporting Information files.

**Funding:** The author(s) received no specific funding for this work.

## Abstract

### Background

Hypertension is a silent killer disease and the global report revealed that half of the world's population lives with undiagnosed hypertension. The problem is expected to be worse in low-income countries such as in Horn of Africa countries. Thus, we planned to determine the trend, burden, and determinates of undiagnosed hypertension in this region and provide conclusive and tangible evidence for interventions.

### Method

Articles were searched on Google, Google Scholar, PubMed/Medline, EMBASE, SCOPUS, and the published articles' reference list. The JBI critical appraisal checklist was used for quality assessment. A sensitivity test and $I^2$ statistics were conducted to evaluate the heterogeneity. The Begg's statistics in the random effect model were done to evaluate the publication bias.

### Result

The pooled prevalence of undiagnosed hypertension in the Horn of Africa was 17% (95% CI: 15%–20%) and it ranges from from 13% in 2006 to 20% in 2023. A trip time to a medical institution of less than 35 minutes (OR = 0.52, 95%CI: 0.35–0.79), no regular exercise (OR = 2.26, 95% CI: 1.54–3.32), age >= 45 years (OR = 2.51, 95% CI: 1.66–3.8), age 35–44 years (OR = 1.88, 95% CI: 1.5–2.37), male (OR = 1.72, 95% CI: 1.34–2.2), poor knowledge (OR = 3.29, 95%CI: 2.39,4.53), normal BMI (OR = 3.84, 95% CI: 2.96–4.98), Overweight (OR = 1.97, 95% CI: 2.96–4.98), poor health seeking (OR = 2.79, 95%CI: 2.01–3.86), low vegetable consumers (OR = 1.99, 95%CI:1.36–2.91), smoking (OR = 1.47, 95%CI: 1.13–1.93), high triglyceride (OR = 1.83, 95%CI:1.33–2.52), chat chewing (OR = 2.18, 95%CI: 1.54–3.09), and alcohol drinking (OR = 1.75, 95%CI: 1.32–2.33) were the determinats of undiagnosed hypertension.

**Competing interests:** The authors have declared that no competing interests exist.

**Abbreviations:** BMI, Body Mass Index; CI, Confidence Interval; OR, Odds Ratio; PRISMA, Prefered Reporting Item for Systematic Review and Meta Analysis; WHO, World Health Organization.

## Conclusion and recommendation

The pooled prevalence of undiagnosed hypertension was low in the Horn of Africa but its trend was increased over time. Individual level variables were identified that affect the undiagnosed hypertension. Therefore, healthy lifestyle is recommended.

## Introduction

Hypertension can be defined by two or more repeated measurements of systolic blood pressure $\geq$ 130mm Hg or $\geq$ 80 for diastolic pressure [1]. Hypertension is a silent killer disease that rarely shows signs and symptoms. Because of the nature of the disease, most people do not seek medical care and go to health institutions when the disease becomes more advanced or complicated [2]. Unmanaged hypertension leads to stroke, premature death, heart disease, chronic kidney disease, and other causes of mortality [3]. Approximately one billion people have developed hypertension around the globe, and if essential intervention is not given, by 2025, hypertension will have increased to 1.5 billion cases [4]. The global report revealed that half of the world's population lives with undiagnosed hypertension [5]. Among the people who are living with hypertension, half of them are unaware of whether they are living with hypertension or not [6]. Overall, increased blood pressure can cause 9 million ischemic heart diseases and 3.5 million strokes each year [7].

Of the global hypertension cases, Africa contributes about 46% [8]. Furthermore, the specific death rate of hypertension in Africa accounts for 25% of the total deaths [9]. Even though the burden of hypertension is high, most African countries cannot afford the health service costs because there are other prioritized health conditions and an inadequate supply of drugs and other medical equipment [10]. Nations around the world, including the Horn of Africa, have been working to meet the target of Sustainable Development Goal Target 3.4 of reducing the premature mortality rate of non-communicable disease by one-third through early prevention and control in 2030 [11].

Undiagnosed hypertension in the Horn of Africa is becoming a major public health agenda. Evidence shows that hypertension has become a hidden epidemic [12, 13]. Undiagnosed hypertension can be attributed to or associated with smoking, an unhealthy diet (for example, high intake of salt, fat, and refined sugar stress), a lack of regular physical activity, a body mass index greater than 25, older age, and alcohol drinking or an unhealthy lifestyle [14, 15]. African countries have been working to reduce the health and economic impact of hypertension, but still, the magnitude of the problem is rising. Furthermore, there is no updated information about the burden, trend, and prevalence of undiagnosed hypertension in the Horn of Africa. Thus, this systematic review and meta-analysis aimed to try to determine the trend, prevalence, and determinate factors of undiagnosed hypertension in the Horn of Africa.

## Methodology

### Design

Though we planned to include all types of study design, all searched articles were conducted using cross-sectional studies and included for analysis for this study. The Preferred Reporting Items for Systematic Reviews and Meta-Analysis Protocols (PRISMA-P 2015 Guidelines) were used [16].

## Searching strategy

All the important published and unpublished papers were searched on Google, Google Scholar, PubMed/Medline, EMBASE, SCOPUS databases, and the published article's reference list from January 23, 2023, to February 30, 2023. The searching mechanism was established using Medical Subject Heading (MeSH) terms by combining the key terms of the title. Again, reviewing the published article reference list was used to identify those papers that were missing on Google, Google Scholar, PubMed/Medline, and EMBASE. Content experts were advised to obtain more papers or to reduce the risk of missing them. Six authors (MCA, MA, MTA, NMD, FM, and TY) searched the articles independently, cross-checking their results. This systematic review and meta-analysis were conducted by extracting studies in the context of the prevalence of undiagnosed hypertension using the following key entry terms. "Undiagnosed hypertension OR high blood pressure AND Ethiopia OR Djibouti OR Eritrea OR Somali OR Uganda OR Sudan OR South Sudan OR Kenya" (**S1 File**).

## Study selection and quality appraisal

The retrieved articles were imported into EndNote X8 software. The duplicated articles were removed after all articles were imported into one database with endnote X8. The title and abstracts of each article were screened by paired authors (MCA and NMD). The discrepancy among the paired authors was solved by discussion and other review teams (MA, FM, DA, TY, and MTA). Articles with full text were considered for systematic and meta-analysis. To assure the quality of the retrieved and included articles, Joana Briggs's Institute critical appraisal checklist for simple prevalence [17] and analytical cross-sectional studies [18] Nine and eight checklist items, respectively, were considered. Articles with a total quality score of more than 50% were labelled as paper-qualified articles, indicating a low risk of bias and articles with total quality score of less than 50% were classified as high risk of bias [18]. During the quality appraisal, these discrepancies were solved by discussion.

## Criteria

**Inclusion criteria.** The authors screened all the papers' titles, abstracts, and full texts. Then:

- Those articles report the outcome of interest, namely the prevalence of undiagnosed hypertension or factors associated with undiagnosed hypertension.

- All cross-sectional, prospective/retrospective cohort studies, case-control studies, and cross-sectional studies were considered.

- Those articles are published in English and published at any time.

- Both published and unpublished articles were incorporated.

- Studies conducted or published at any time were included in the study.

**Exclusion criteria.** The authors critically, comprehensively, independently, and blindly screen the abstracts and the full texts. After this extensive screening, the authors excluded the following articles:

- Articles that did not report the outcome of interest, namely the prevalence of undiagnosed hypertension or factors associated with undiagnosed hypertension, were excluded.

- Case reports and case series studies were excluded.

- Articles that didn't have full text were excluded because of difficulties in quality assessment.

### Outcomes of the study

The primary outcome of the intended systematic and meta-analysis was the prevalence, trend of undiagnosed hypertension and associated factors in the Horn of Africa.

### Operational definition

**Undiagnosed hypertension:** was defined as systolic blood pressure of 140 mmHg and above and/or diastolic blood pressure of 90 mmHg and above, without previous history or anti-hypertensive treatment during the survey diagnosis [19].

### Data extraction

Next to the exhaustive data searching and screening, eligible articles were extracted using a data extraction format that includes the name of the first author, publication year, study area or region, study design, tool, quality of the articles, number of samples, number of undiagnosed hypertension, proportions, standard error of the proportion, odds ratio for each independent variable, confidence interval of the odds ratio, and standard error of each odds ratio. These were extracted from each study using a Microsoft Excel spreadsheet. To extract the data, the JBI tool was adopted. The three paired authors extracted the data independently using the same standard format and then cross-checked the consistency of the extracted data. The cross-checking of the extracted data includes the consistency of all the data extracted from the three groups of paired authors. For example, the consistency of the quality score of each article, the number of samples for each article, the proportions of undiagnosed hypertension in each article, the standard error of the proportion, the odds ratio for each independent variable, the confidence interval of the odds ratio, and the standard error of each odds ratio. The discrepancy, such as the inconsistency in the quality assessment of the paper, was solved by discussion between the three paired teams. The majority agreement was not solved by group consensus.

### Data analysis

Data from the eligible articles was entered into Microsoft Excel and then exported to Stata 14 for further data analysis. The $I^2$ statistics, or the index of heterogeneity, was used to quantify the amount of heterogeneity between the included studies [20]. Values of $I^2$ of 25%, 50%, and 75% were considered low, medium, and high heterogeneity, respectively [21]. The significance of the heterogeneity between the studies was evaluated by a p-value of $I^2$ less than 0.05. Sensitivity analysis and subgroup analysis using the study setting and regions were conducted to evaluate and manage the heterogeneity. The visual funnel plot test (visual method) and Egger's statistics (objective/statistical technique) in the random effect model were done to evaluate the publication bias or small study effect. Additionally, to manage the heterogeneity between the studies, a random effect model was used. The random effect model assumes that the true treatment effects in the individual studies may be different from each other. It is not only variation within the study but also variation across the study. To consider this variation, we used the random effect model. But the fixed effect model should be used when there is little evidence of heterogeneity. The effect size was reported by OR with its 95% confidence level to estimate the statistical association between undiagnosed hypertension and associated factors. The prevalence of undiagnosed hypertension and other factors was reported in both the text and the forest plot.

## Results

Out of 247,774 searched articles, 243,987 were excluded because of duplicated articles. About 2519 records were screened, and 264 of them were assessed for eligibility. Finally, 43 potential studies have been included for systematic review and meta-analysis, as summarized in the PRISMA flow diagram (**Fig 1**).

### Characteristics of included studies

All included studies were conducted using a cross-sectional study design. Of the total included studies, the majority, 28 (65.1%) were conducted in an urban setting. The total number of participants included or used for analysis was 64,156, and the minimum and maximum sample sizes used in the studies were 170 [22] and 17,777 [23] respectively (**Table 1**).

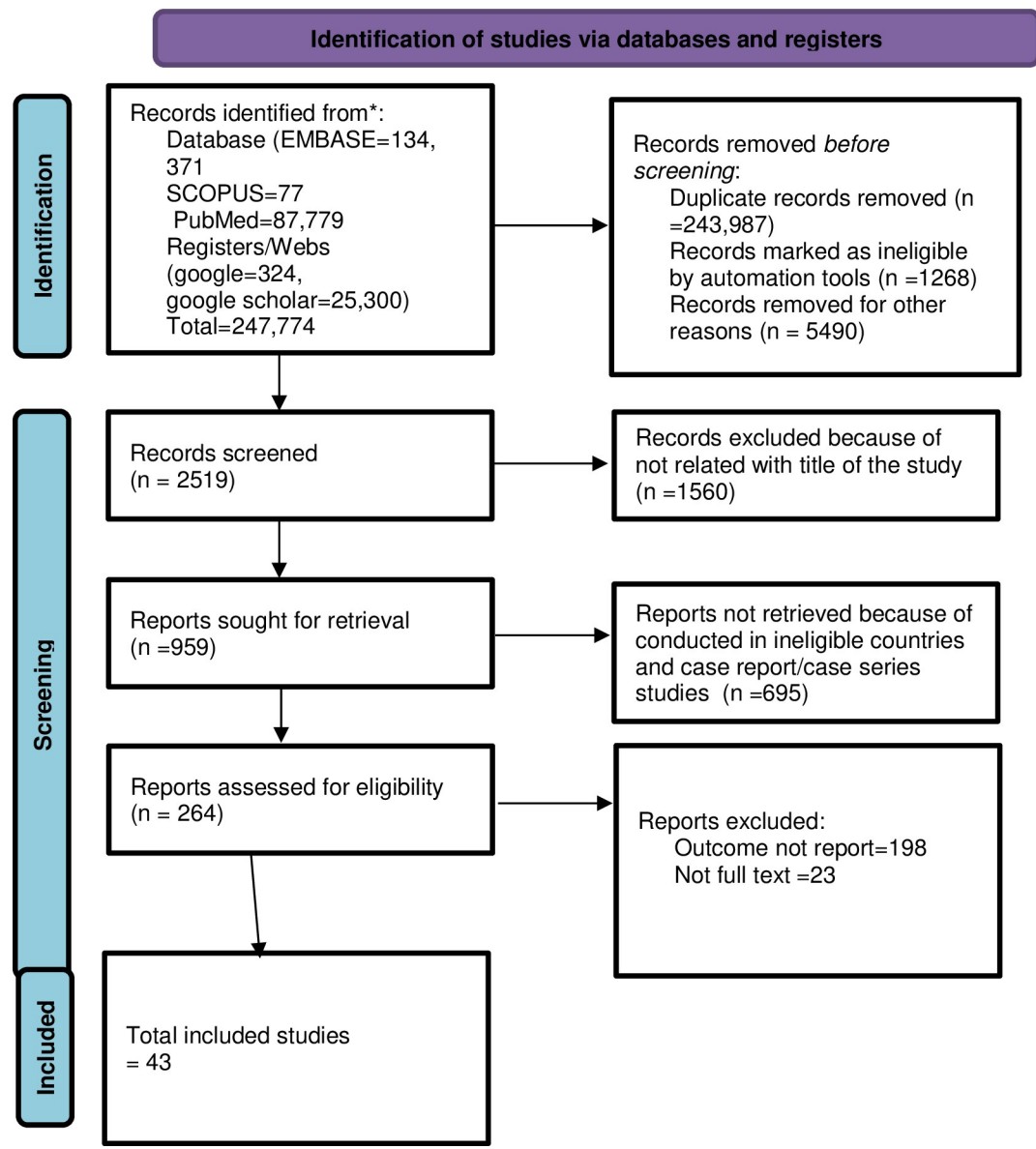

**Fig 1. PRISMA flow diagram of study selection for the trend, prevalence and determinants of undiagnosed hypertension in the Horn of Africa, 2023.**

**Table 1. Characteristics of the included studies for the study of undiagnosed hypertension in the Horn of Africa.**

| S. No | Author | Publication Year | Country | Area/residence | Quality | Study setting | Sample size | Prevalence of undiagnosed hypertension |
|---|---|---|---|---|---|---|---|---|
| 1 | Dejenie M *et al* [24] | 2020 | Ethiopia | urban | low risk(60%) | Institutional | 513 | 24.80% |
| 2 | Mengistu Y [25] | 2022 | Ethiopia | urban | low risk(70%) | Institutional | 191 | 25.10% |
| 3 | Essa E *et al* [26] | 2022 | Ethiopia | urban | low risk (67%) | Institutional | 600 | 12.70% |
| 4 | Getachew F *et al* [27] | 2018 | Ethiopia | urban | low risk (55.6%) | Institutional | 422 | 13.25% |
| 5 | Wachamo D *et al* [28] | 2019 | Ethiopia | urban | low risk(80%) | Institutional | 383 | 12.30% |
| 6 | Dereje N *et al* [29] | 2020 | Ethiopia | urban | low risk(76%) | Institutional | 634 | 6.90% |
| 7 | Elias S, Dadi TK [30] | 2022 | Ethiopia | urban | low risk (66.7%) | Institutional | 738 | 14.80% |
| 8 | Teshome DF *et al* [31] | 2023 | Ethiopia | rural | low risk (55.6%) | Community | 2436 | 84.00% |
| 9 | Regea F *etal* [32] | 2022 | Ethiopia | rural | low risk (66.7%) | Institutional | 605 | 14.60% |
| 10 | Mogas SB *et al* [15] | 2021 | Ethiopia | urban | low risk (55.6%) | Institutional | 915 | 21.10% |
| 11 | Demamu H *et al* [33] | 2021 | Ethiopia | rural | low risk (66.7%) | Community | 574 | 45.30% |
| 12 | Asemu MM *et al* [34] | 2021 | Ethiopia | urban | low risk (77.8%) | Community | 3560 | 19.29% |
| 13 | Tesfaye T [35] | 2021 | Ethiopia | urban | low risk (77.8%) | Institutional | 584 | 24.30% |
| 14 | Tesfaye TD [36] | 2019 | Ethiopia | mixed urban and rural | low risk (88.9%) | Community | 1405 | 5.33% |
| 15 | Bonsa F *et al* [37] | 2014 | Ethiopia | urban | low risk (77.8%) | Community | 396 | 9.34% |
| 16 | Gelassa FR *et al* [38] | 2022 | Ethiopia | rural | low risk (77.8%) | Community | 605 | 21.32% |
| 17 | Roba HS *et al* [39] | 2019 | Ethiopia | urban | low risk (77.8%) | Community | 903 | 12.60% |
| 18 | Helelo TP *et al* [40] | 2014 | Ethiopia | urban | low risk (66.7%) | Community | 518 | 9.10% |
| 19 | Bekele A *et al* [41] | 2017 | Ethiopia | mixed urban and rural | low risk (55.6%) | Community | 9788 | 15.14% |
| 20 | Asresahegn H, [42] | 2017 | Ethiopia | urban | low risk (77.8%) | Community | 487 | 18.10% |
| 21 | Yadecha B *et al* [43] | 2020 | Ethiopia | urban | low risk (66.7%) | Community | 471 | 24.84% |
| 22 | Awoke A.*et al* [44] | 2012 | Ethiopia | urban | low risk (77.8%) | Community | 679 | 10.60% |
| 23 | Asfaw LS *et al* [45] | 2018 | Ethiopia | urban | low risk (66.7%) | Community | 524 | 5.40% |
| 24 | Paulose T *et al* [46] | 2022 | Ethiopia | urban | low risk (66.7%) | Community | 612 | 9.00% |
| 25 | Essayas A *et al* [47] | 2018 | Ethiopia | urban | low risk (55.6%) | Institutional | 618 | 7.3% |
| 26 | Geleta GT *et al* [48] | 2019 | Ethiopia | urban | low risk (55.6%) | Community | 705 | 16.59% |
| 27 | Abebe SM *et al* [49] | 2015 | Ethiopia | mixed urban and rural | low risk (88.9%) | Institutional | 2141 | 23.31% |
| 28 | Shukuri A *et al* [50] | 2019 | Ethiopia | rural | low risk (55.6%) | Community | 401 | 35.91% |

*(Continued)*

**Table 1.** (Continued)

| S. No | Author | Publication Year | Country | Area/residence | Quality | Study setting | Sample size | Prevalence of undiagnosed hypertension |
|-------|--------|------------------|---------|----------------|---------|---------------|-------------|----------------------------------------|
| 29 | Badego B *et al* [51] | 2020 | Ethiopia | urban | low risk (77.8%) | Community | 575 | 3.82% |
| 30 | Kebede B *et al* [52] | 2020 | Ethiopia | urban | low risk (66.7%) | Community | 784 | 31.76% |
| 31 | Gudina EK *et al* [53] | 2013 | Ethiopia | mixed urban and rural | low risk (88.9%) | Community | 734 | 8.58% |
| 32 | Bayray A *et al* [54] | 2018 | Ethiopia | urban | low risk(66.75) | Community | 1523 | 5.31% |
| 33 | Kumma WP *et al* [55] | 2021 | Ethiopia | urban | low risk (88.9%) | Community | 2483 | 9.34% |
| 34 | Belay DG *et al* [56] | 2022 | Ethiopia | mixed urban and rural | low risk (77.8%) | Community | 432 | 28.70% |
| 35 | Berhe DA *et al* [57] | 2020 | Ethiopia | urban | low risk (55.6%) | Institusional | 414 | 17.39% |
| 36 | Ayalew. TL *et al* [58] | 2022 | Ethiopia | Urban | Low risk (66.7%) | Community | 644 | 28.8% |
| 37 | J Mufunda *et al* [22] | 2006 | Eritrea | Mixed urban and rural | Low risk (55.6%) | Community | 2352 | 12.7% |
| 38 | S.O. Bushara *et al* [59] | 2015 | Sudan | Rural | Low risk (66.7%) | Community | 1099 | 38.2% |
| 39 | Sawsan A. *et al* [60] | 2023 | Sudan | Mixed urban and rural | Low risk (55.6%) | Community | 464 | 10.1% |
| 40 | Sufan K *et al* [61] | 2013 | Sudan | Urban | Low risk (66.7%) | Institusional | 170 | 38.2% |
| 41 | Supa Pengpid *et al.*[62] | 2022 | Sudan | Mixed urban rural | Low risk (88.9%) | Community | 2057 | 26.2% |
| 42 | Kilama *et al* [63] | 2023 | Uganda | Mixed urban rural | Low risk (66.7%) | Community | 240 | 16.7% |
| 43 | Usnish Majumdar *et al* [23] | 2022 | Uganda | Urban | Low risk (66.7%) | Institusional | 17,777 | 19% |

## The pooled prevalence of undiagnosed hypertension in the Horn of Africa

In the random effects model, the pooled prevalence of undiagnosed hypertension in the Horn of Africa was 17% (95% CI, 15%–20%), and the heterogeneity among the studies was statistically significant ($I^2 = 98.42\%$, Pvalue $< 0.001$) (**Fig 2**).

## Assessment of publication bias

Egger's statistical test evidenced that there was no publication bias among the included studies ($\beta = 7.947$, P-value $= 0.158$) (**Fig 3**).

## Handling heterogeneity

In the random effect model of pooled prevalence, statistically significant heterogeneity was detected. To solve this problem, sensitivity analysis and subgroup analysis were performed.

## Sensitivity analysis

To manage the influence of a single study in meta-analysis estimation (to assess the heterogeneity), the random effects model and sensitivity analysis were conducted, and no study excessively influenced the overall pooled prevalence of undiagnosed hypertension (**(S1 Fig) in S2 File**).

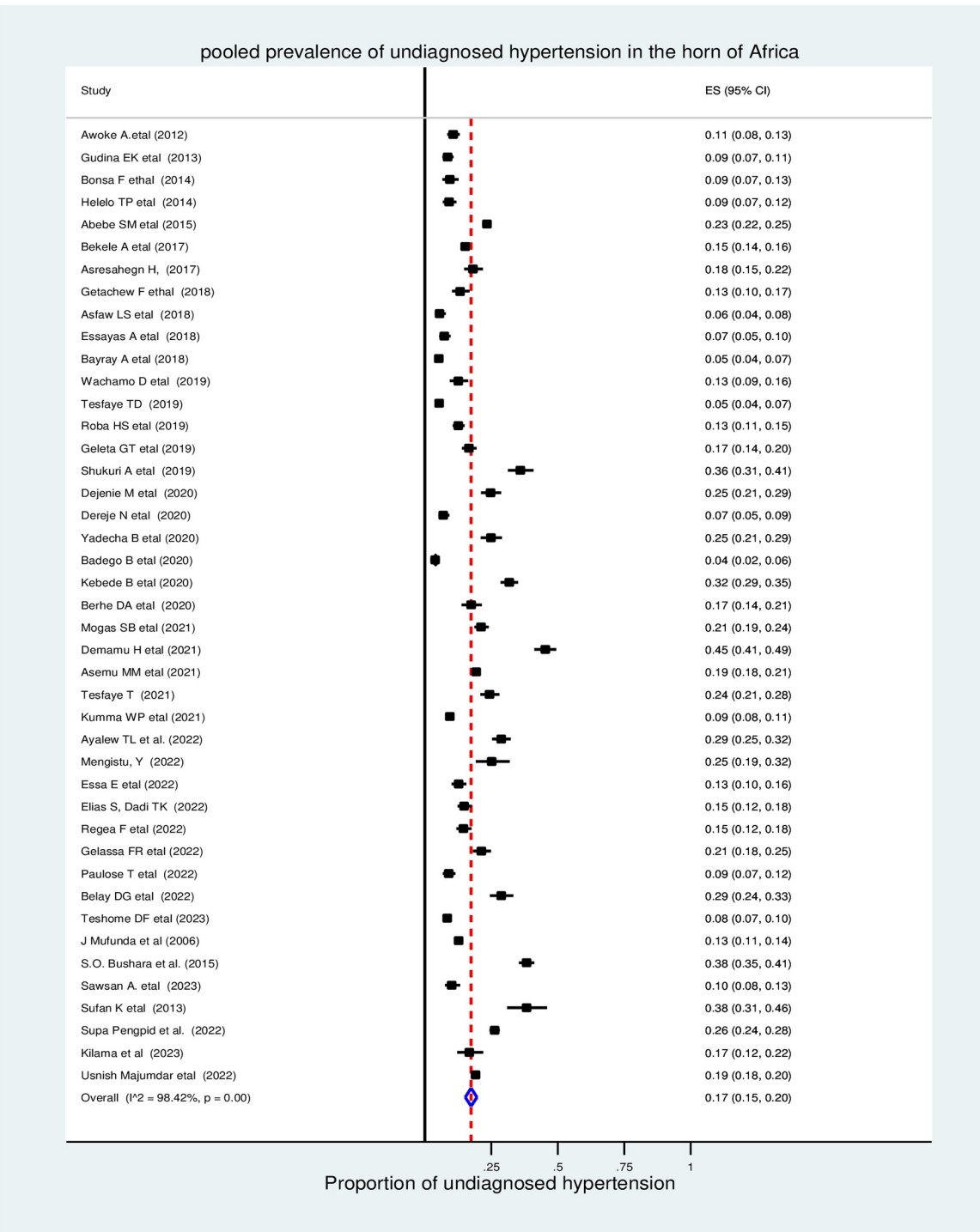

**Fig 2. The pooled prevalence of undiagnosed hypertension in the Horn of Africa, 2023.**

## Subgroup analysis by country

In the subgroup analysis by country, the highest prevalence of undiagnosed hypertension was observed in Sudan (28%; 95% CI: 16%–40%) **(Fig 4)**.

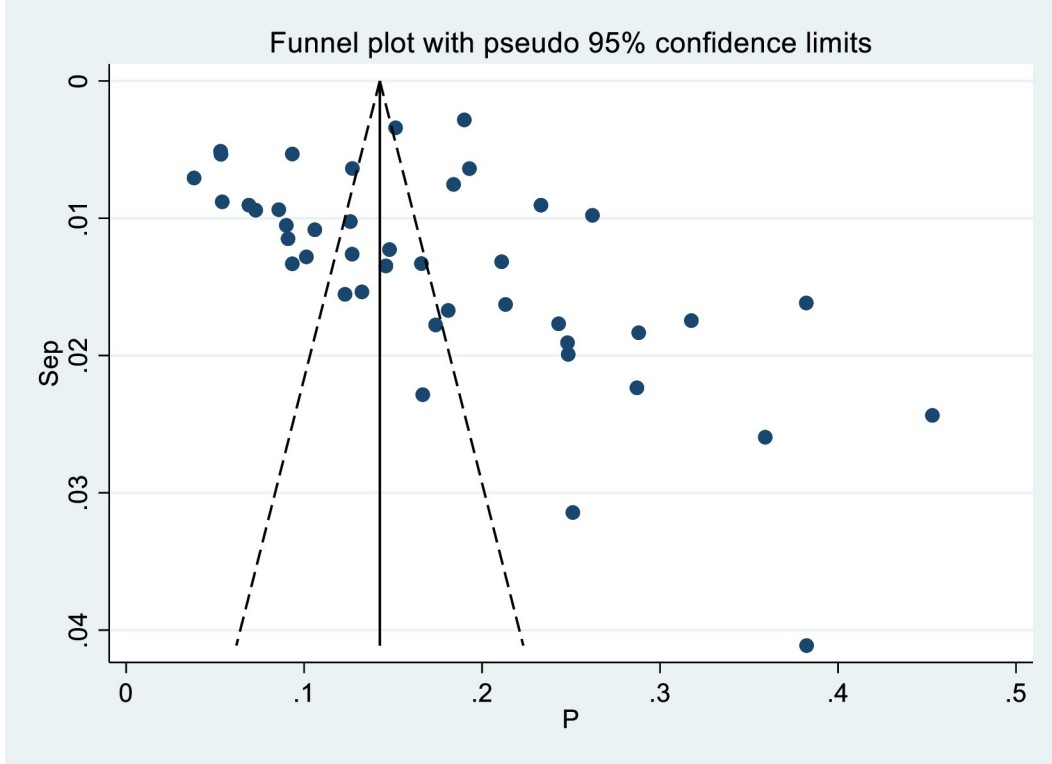

```
Number of studies =   43                          Root MSE    =    7.947

    Std_Eff │ Coefficient  Std. err.      t    P>|t|     [95% conf. interval]

      slope │   .1202394    .0185998    6.46   0.000     .0826764    .1578024
       bias │   3.157316    2.194321    1.44   0.158    -1.274205    7.588836

Test of H0: no small-study effects            P = 0.158
```

**Fig 3. Funnel plot of undiagnosed hypertension in the Horn of Africa.**

### Subgroup analysis by area/residence

Subgroup analysis was also carried out in the area/residence as a source of heterogeneity. The highest prevalence of undiagnosed hypertension was observed in rural areas (27%; 95% CI: 14%–40%) (**Fig 5**).

### Trends of undiagnosed hypertension in the Horn of Africa

The trend of undiagnosed hypertension in the Horn of Africa increased over time, from 13% in 2006 to 20% in 2023 and it was statistically significant (**Fig 6**).

### Factors associated with undiagnosed hypertension

As briefly described in Table 2, sex, age, BMI, chat chewing, alcohol consumption, regular exercise, knowledge about hypertension, health-seeking behavior, vegetable consumption,

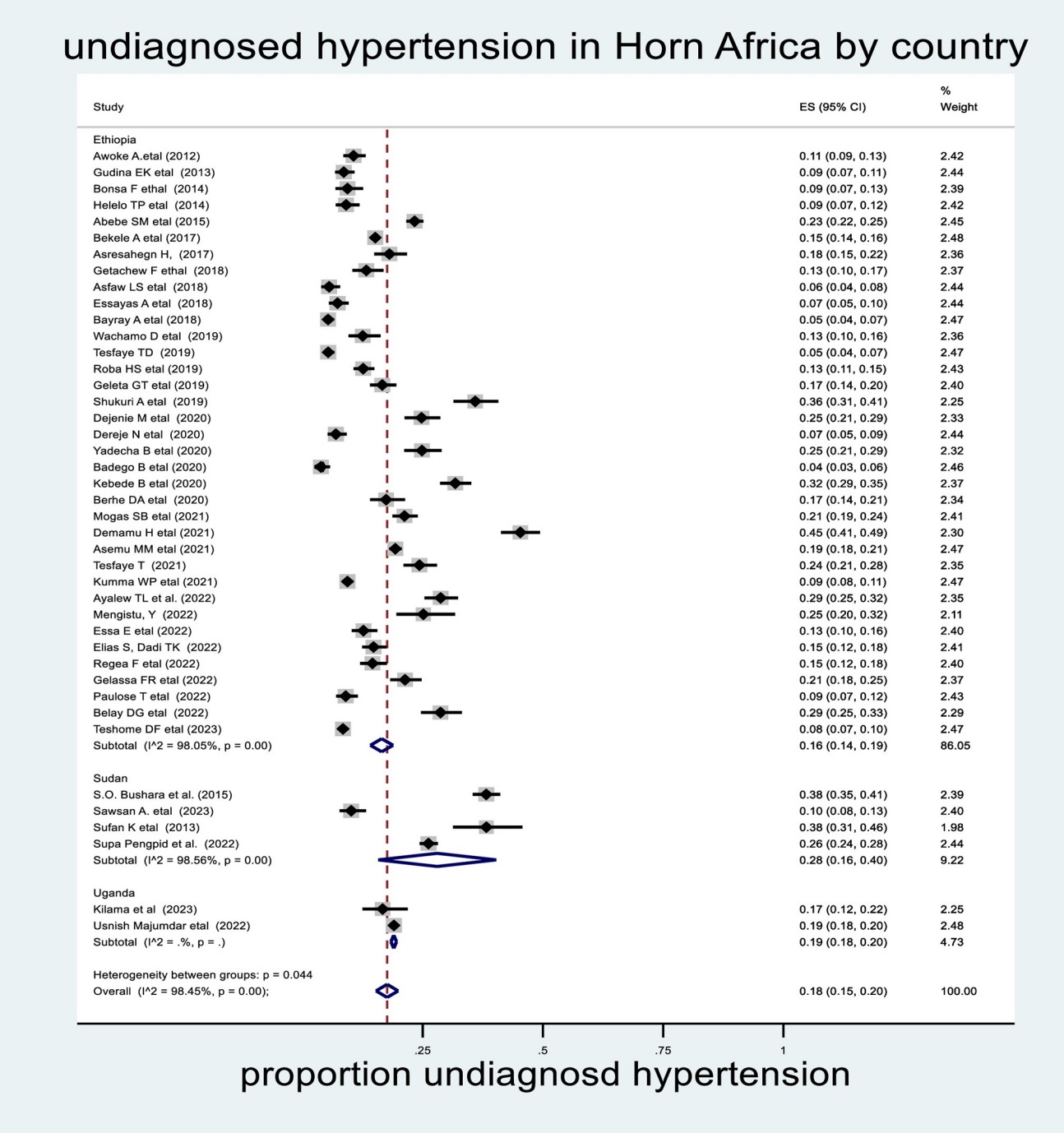

**Fig 4. The pooled prevalence of undiagnosed hypertension in the Horn of Africa.** By country, 2023.

level of triglyceride, and time to reach the health facilities were statistically significant factors for undiagnosed hypertension in the Horn of Africa. From the random effect model estimate, the pooled odds of undiagnosed hypertension among males were 1.81 times higher than those of females (OR = 1.81, 95% CI: 1.42–2.3), with $I^2$ = 72.5% and a P-value of 0.001. Egger's test indicates that there was no small study effect (P-value = 0.454), and the random effects model

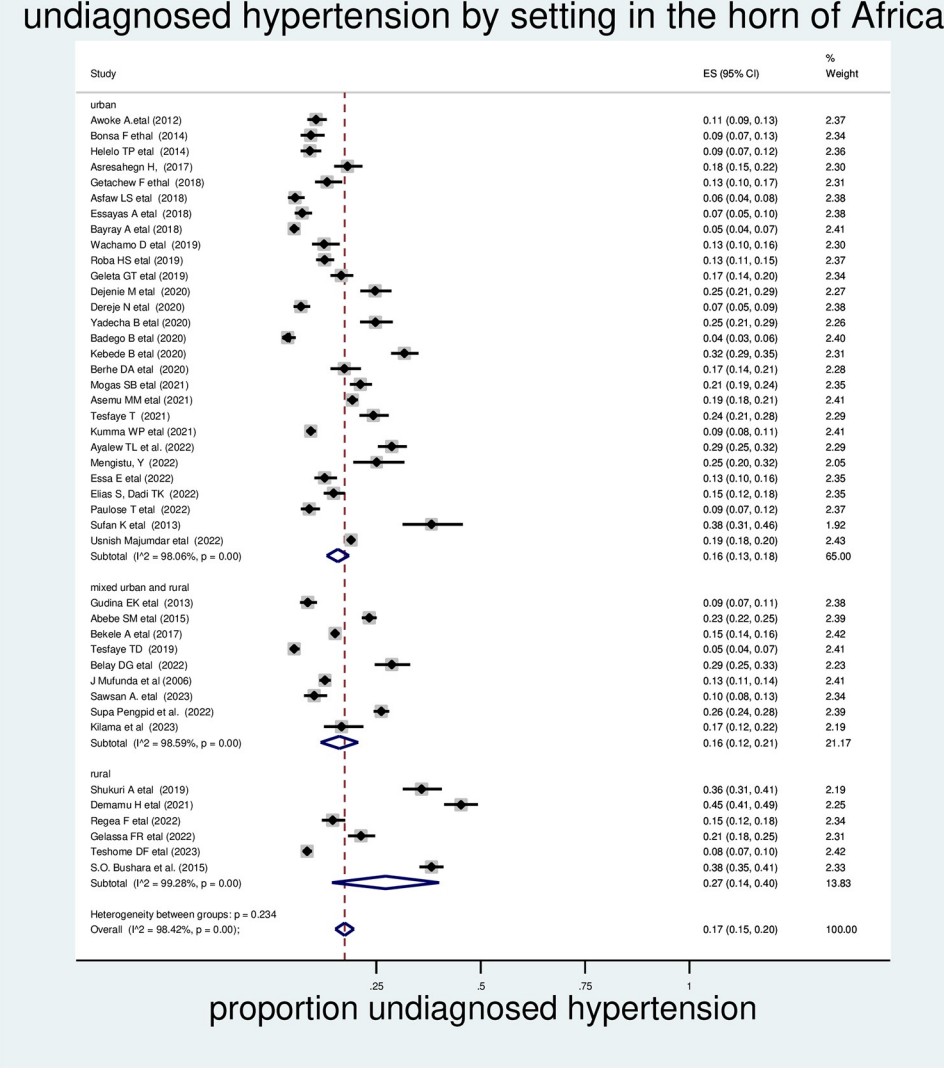

**Fig 5. The pooled prevalence of undiagnosed hypertension in the Horn of Africa by study setting, 2023.**

reported no single study that excessively influences the effect size of this factor (**Table 2, (S5 Fig) in S3 File**).

Another significant factor for undiagnosed hypertension was age, and thus age $>= 45$ years and 35–44 years was 2.49 (OR = 2.49, 95% CI: 2.09–2.97), ($I^2 = 0\%$, P-value = 0.72), and 1.88 (OR = 1.88, 95% CI: 1.5–2.37, $I^2 = 22.5\%$, P-value = 0.266) times higher risk for undiagnosed hypertension than 18–34 years, respectively. The Egger's test indicates that there was no small study effect in age $>= 45$ years (P-value = 0.444) and 35–44 years (P = 0.502). This means that in the random effects model, there was no single study that excessively influenced the effect size of this factor (**Table 2, (S3 Fig) in S3 File**). The pooled effect size of undiagnosed hypertension among those who had no regular exercise was 2.2 times more likely than the counterpart (OR = 2.26, 95% CI: 1.54–3.32, $I^2 = 64.3\%$, P-value = 0.023). Egger's test indicates that there was no small study effect (P-value = 0.524), and in the random effect model, there was no single study that excessively influenced the effect size of this factor (**Table 2, (S14 Fig) in S3 File**).

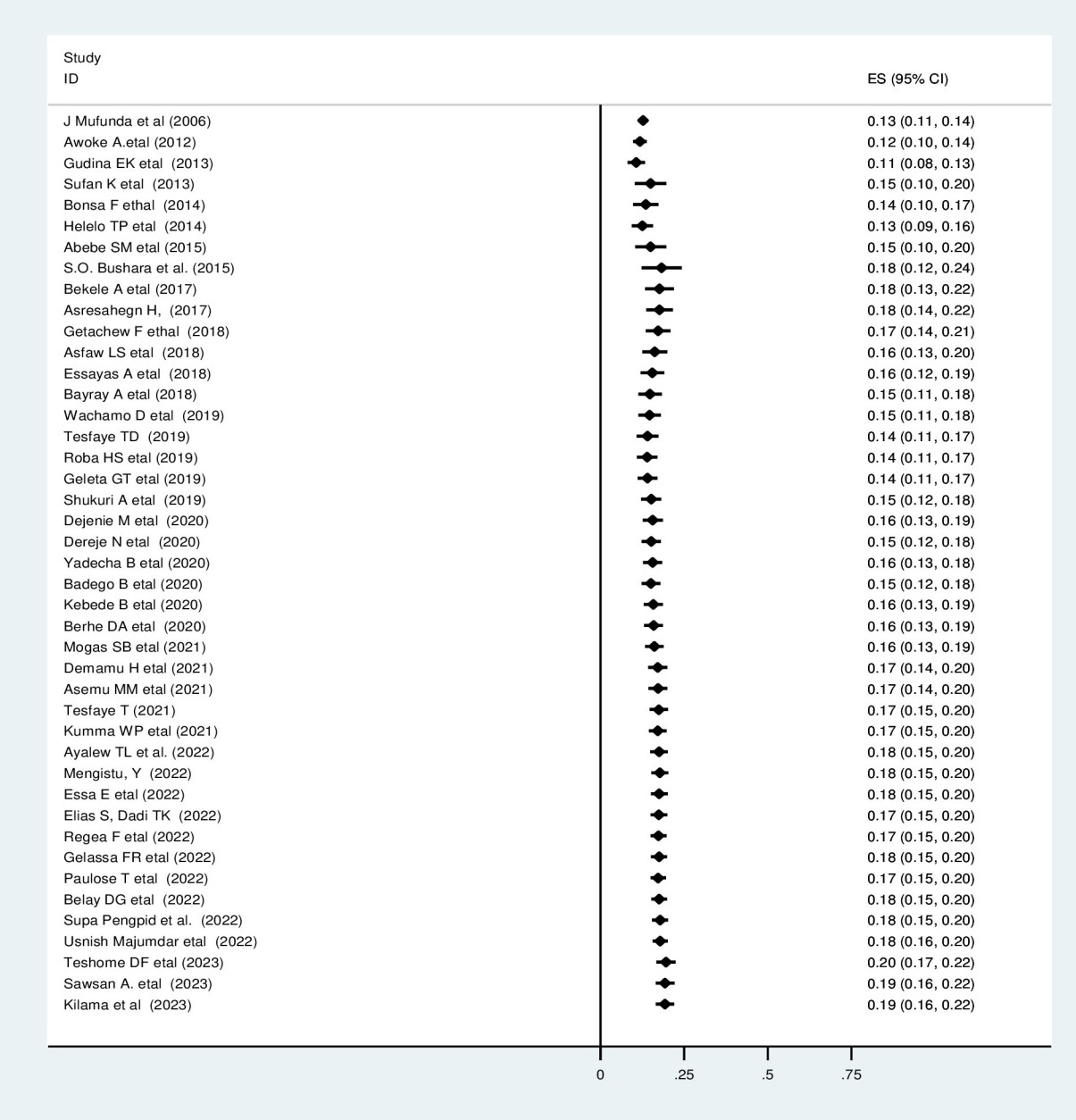

**Fig 6. The trend of undiagnosed hypertension in the Horn of Africa.**

The pooled odds of undiagnosed hypertension in a trip time to a medical institution of less than 35 minutes were reduced by 48% compared to those above 35 minutes to reach the health facilities (OR = 0.52, 95% CI: 0.35–0.79, $I^2$ = 88.6%, P-value = 0.003). Egger's test indicates that there was no small study effect (P-value = 0.245), and in the random effects model, there was no single study that excessively influenced the effect size of this factor (**Table 2, (S10 Fig) in S3 File).**

Regarding knowledge of hypertension, adults who had poor knowledge about hypertension were 3.29 times more likely to have undiagnosed hypertension than those with good knowledge about hypertension (OR = 3.29, CI: 2.39, 4.53), with $I^2$ = 43.7% and a P-value of 0.137.

**Table 2. Summary of factors associated with undiagnosed hypertension in the Horn of Africa., 2023.**

| Variable | | OR(95%CI) | Heterogeneity (I$^2$,P-value) | Total studies | Sample size |
|---|---|---|---|---|---|
| Sex | Male | 1.81(1.42, 2.3)* | 72.5%%, 0.001 | 8 | 3914 |
| | Female | 1 | 1 | | |
| Monthly income | <2000 ETB | 0.71(0.51,1.0) | 0%, 0.331 | 2 | 1174 |
| | >2000ETB | 1 | 1 | | |
| Age | 18–34 | 1 | 1 | | |
| | 35–44 | 1.88(1.5,2.37)* | 22.5%, 0.266 | 6 | 5353 |
| | > = 45 | 2.49(2.09,2.97)* | 0%, 0.702 | 9 | 7841 |
| Regular exercise | Yes | 1 | 1 | | |
| | No | 2.26(1.54,3.32)* | 64.3%, 0.023 | 5 | 2846 |
| Excess salt consumption | Yes | 1.19(0.58, 2.47) | 87.4%, 0.005 | 2 | 990 |
| | No | 1 | 1 | | |
| Distance | <35min | 0.52(0.35,0.79)* | 88.6%, 0.003 | 2 | 1210 |
| | 35> = min | 1 | 1 | | |
| Knowledge | Good | 1 | 1 | | |
| | Poor | 3.29(2.39, 4.53)* | 45.7%, 0.137 | 4 | 2297 |
| BMI | Normal | 3.84(2.96,4.98)* | 96.1%,<0.001 | 4 | 2430 |
| | Overweight | 2.11(1.57,2.87)* | 76.7%,<0.001 | 7 | 5695 |
| | Underweight | 1 | 1 | | |
| Family history of hypertension | Yes | 1.13(0.84,1.5) | 88%, <0.001 | 6 | 5374 |
| | No | 1 | 1 | | |
| | Yes | 2.16(1.47,3.16)* | 94.8%,<0.001 | 2 | 1343 |
| | No | 1 | 1 | | |
| Occupation | Housewife | 1.84(0.97,3.48) | 88.8%,0.008 | 2 | 1022 |
| | Merchant | 0.819(0.43,1.53) | 0%, 0.673 | 2 | 1022 |
| | Unemployed | 1.14(0.65,1.97) | 14.4%,0.32 | 4 | 2273 |
| | Government employed | 1 | 1 | | |
| Health seeking | Good | 1 | 1 | | |
| | Poor | 2.79(2.01, 3.86)* | 0%, 0.646 | 4 | 2174 |
| Vegetable consumption | Low | 1.99(1.36,2.91)* | 61%,0.036 | 5 | 2663 |
| | Not-consume | 1.23(0.88,1.83) | 58.2%,0.091 | 3 | 1540 |
| | Normal | 1 | 1 | | |
| Smoking | Yes | 1.47(1.13,1.93)* | 91.5%,<0.001 | 5 | 3166 |
| | No | 1 | 1 | | |
| triglyceride level | High | 1.58(1.32,1.89)* | 35.1%, 0.202 | 4 | 4221 |
| | Normal | 1 | 1 | | |
| Chat chewing | Yes | 2.18(1.54,3.09)* | 89.7%,<0.001 | 3 | 1987 |
| | No | 1 | 1 | | |
| Alcohol drinking | Yes | 1.75(1.32,2.33) | 77.2%,0.002 | 5 | 5023 |
| | No | 1 | 1 | | |

Egger's test indicates that there was no small study effect (P-value = 0.95), and thus there was no single study that excessively influenced the effect size of this factor (**Table 2, (S12 Fig) in S3 File**).

Again, the odds of undiagnosed hypertension among normal BMI and overweight were 3.84 (OR = 384, 95% CI: 2.96–4.98, I$^2$ = 96.1%, P-value = <0.001) and 2.11 (OR = 2.11, CI: 1.57–2.83, I$^2$ = 76.7%, P-value = <0.001) times higher than underweight. Egger's test indicates that there was no small study effect for normal BMI (P-value = 0.549), for overweight (P-

value = 0.139), and thus there was no single study that excessively influences the effect size of this factor (**Table 2**, **(S6 Fig)** in **S3 File**).

Having poor health-seeking behaviour was 2.79 times more likely to have undiagnosed hypertension than having good health-seeking behaviour (OR = 2.79, CI: 2.01–3.86, $I^2$ = 0%, P-value = 0.646). Egger's test also indicates that there was no small study effect (P-value = 0.236), and thus there was no single study that excessively influenced the pooled estimate of this factor (**Table 2**, **(S13 Fig)** in **S3 File**).

Another factor that influences undiagnosed hypertension is vegetable consumption. Thus, the odds of undiagnosed hypertension among low-vegetable consumers were 1.99 times higher than those of normal consumers (OR = 1.99, 95% CI: 1.36–2.91, $I^2$ = 61%, P-value = 0.036). Egger's test indicates that there was no small study effect (P-value = 0.928), and thus there was no single study that excessively influenced the effect size of this factor (**Table 2**, **(S4 Fig)** in **S3 File**).

Additionally, the odds of undiagnosed hypertension among smokers were 1.47 times higher than nonsmokers (OR = 1.47, 95% CI: 1.13–1.93, $I^2$ = 91.5%, P-value<0.001). Egger's test indicates that there was no small study effect (P-value = 0.715), and thus there was no single study that excessively influenced the effect size of this factor (**Table 2**, **(S7 Fig)** in **S3 File**). Having a high level of triglyceride was 1.58 times more likely to put you at risk for undiagnosed hypertension than a normal level of triglyceride (OR = 1.58, 95% CI: 1.31–1.89, $I^2$ = 35.1%, P-value = 0.202). Egger's test showed that there was no small study effect (P-value = 0.795), and thus there was no single study that excessively influenced the effect size of this factor (**Table 2**, **(S11 Fig)** in **S3 File**).

The odds of undiagnosed hypertension among chat chewers were 2.18 times higher than their counterparts (OR = 2.18, CI: 1.54–3.09, $I^2$ = 89.7%, P-value<0.001). Egger's test indicates that there was no small study effect (P-value = 0.836), and thus there was no single study that excessively influenced the effect size of this factor (**Table 2**, **(S9 Fig)** in **S3 File**).

On the other hand, the odds of undiagnosed hypertension among alcohol drinkers were 1.75 times higher than their counterparts (OR = 1.75, 95% CI: 1.32–2.33, $I^2$ = 77.2%, P-value = 0.002). Egger's test also showed that there was no small study effect (P-value = 0.743), and thus there was no single study that excessively influenced the effect size of this factor (**Table 2**, **(S8 Fig)** in **S3 File**).

## Discussion

In this study, an attempt has been made to assess the trend, burden, and determinate factors of undiagnosed hypertension in the Horn of Africa. Thus, the pooled prevalence of undiagnosed hypertension in the Horn of Africa was 17% (95% CI: 15–20), distance takes <35 min, no regular exercise > = 45 years, 35–44 years, male, good knowledge, normal BMI, overweight, poor health seeking, low vegetable consumers, smokers, high triglyceride, chat chewing, and alcohol drinking were the factors associated with undiagnosed hypertension. The pooled prevalence of undiagnosed hypertension in the current study was lower than a systematic and meta-analysis study conducted in Sub-Saharan Africa (30%), Nigeria. (28.9%), India (29.8%), Pakistan (26.34%), and a study in Nepal (25.1%) [14, 64–68]. This may be because of the difference in the time of study and the difference in the study setting. The other possible reason may be that most of the studies conducted and included in the current study were among the urban population. This segment of the population has better health-seeking and check-up behaviour than the rural population. In the subgroup analysis by study setting and regions, the pooled prevalence of undiagnosed hypertension was higher in rural areas, which was 25% (95% CI: 12–38). The possible reason for this difference may be that the rural population has a lower educational

status, a lack of access to health facilities, and a lack of access to the media. In return, they may have low health-seeking or health check-up behaviour [69].

This study also identifies the associated factors of undiagnosed hypertension in the Horn of Africa, and thus males were more likely to have undiagnosed hypertension than females. This finding was supported by a study done in the United States of America [70]. The possible reason for this finding may be that most of the time, women have the chance to receive various maternal health services. For example, family planning delivery, antenatal care services, and immunization services provide a good opportunity for understanding their health status [24].

The odds of undiagnosed hypertension in ages 35–44 and $>= 45$ years were higher than in ages 18–34 years. This finding is supported by studies done in Nepal [71], Uganda [72] and Benin [73] This is due to arterial stiffness increasing as age increases. In return, it increases blood pressure, and parallelly, undiagnosed hypertension is more common in older people [71].

Physical exercise was also a significant factor for undiagnosed hypertension, and, thus, adults with no regular exercise were at higher risk of undiagnosed hypertension than regular physical exercise. This is because physical activity and/or exercise have been shown to delay the development of hypertension [74]. The other possible reason might be associated with those individuals who live a sedentary life, such as those who have no exercise and are also not seeking medical care.

This study also revealed that the distance from health facilities to home affects undiagnosed hypertension, and thus traveling time <35 minutes reduces the risk of undiagnosed hypertension. This may be because those who are at nearby health facilities access the health service more easily than those who are far away.

Poor knowledge about hypertension is another risk factor for undiagnosed hypertension. The finding was supported by the studies conducted in Rwanda and Cracow [5, 75]. This might be because adults who have low knowledge about hypertension have a low level of screening for hypertension.

Another associated factor for undiagnosed hypertension in the Horn of Africa was BMI. Normal BMI and overweight increase the risk of undiagnosed hypertension more than underweight individuals. This is supported by studies conducted in sub-Saharan African countries [76] and national study in Burkina Faso [77]. This might be because of a 1 kg/m$^2$ increase in BMI; the odds of hypertension also increased as compared to underweight (risk of undiagnosed hypertension = obese/overweight $>$ normal $>$ underweight) [78]. Alternatively, as the BMI increases by 1 kg/m$^2$, there is an increase in the risk of undiagnosed hypertension. In this case, a normal BMI is associated with a associated with a higher risk of undiagnosed hypertension as compared to an underweight (low BMI) by holding other factors constant. Again, those individuals who have a normal BMI may have a poor belief that they have a low risk of developing hypertension, which makes them less likely to visit a health institution to measure their blood pressure. Alternatively, their subjective perception of the lower risk of acquiring hypertension may prevent them from measuring their blood pressure. Even during the selective/risk group screening, these individuals may be missed.

In this study, adults with low health-seeking behaviour were more likely to have undiagnosed hypertension than those with good health-seeking behaviour. This might be because adults with low health-seeking behaviour have less chance to visit health institutions, and, as a result, they do not know their hypertension status [79].

This study also showed that chat chewers are more likely to have undiagnosed hypertension. This is supported by a systematic and meta-analytic study in Ethiopia [80]. This is because chat (Khat) contains cathinone, cathine, and amphetamine [81] and this leads to

elevated blood pressure [82, 83]. Another possible response might be that adults living an unhealthy life give less attention to early health seeking and screening [84].

In the current study, the pooled effect of alcohol drinking was the risk factor for undiagnosed hypertension. This is due to alcohol consumption negatively affecting the central nervous system, which increases cardiac output and affects the peripheral vascular effect, which leads to elevated blood pressure [85]. Another possible reason might be that adults living an unhealthy lifestyle are given less attention to early health-seeking and screening [84].

Another finding in the current study was that adults with low consumption of vegetables were more likely to be at risk for undiagnosed hypertension. This is because vegetables and fruit are very essential food components of a healthy diet that can prevent a variety of cardiovascular diseases, including hypertension, and WHO recommends these food items to enhance the health of the community [86].

Additionally, the current study reported that alcohol drinkers had a higher risk of undiagnosed hypertension than non-alcohol drinkers. The finding is consistent with studies conducted in Tanzania [87] and China [88, 89]. This might be because adults with an unhealthy lifestyle, including alcohol drinking, have less health-care-seeking behavior [84].

Furthermore, the odds of undiagnosed hypertension were higher among adults with a high triglyceride level than a normal level of triglyceride. This might be because high triglycerides are often a sign of other conditions that increase the risk of heart disease, hypertension, and other chronic diseases, and such adults are comorbid. As a result, they seek medical care, have frequent contact with health professionals, and know their blood pressure status. The strength of the study was the involvement of many experts (data extraction by six researchers) during the searching, screening, appraisal, and data extraction processes. Research helped to make the evidence more comprehensive and valid. Furthermore, the data report adheres to guidelines outlined in the PRISMA P-2015 statement protocol. However, some studies that categorize the variables differently and are

difficult to manage were omitted. Additionally, hetrogenicity between the studies and majority of the study were in Ethiopia which may causes the risk of unreperentative for the Horn Africa.

## Conclusion

The pooled prevalence of undiagnosed hypertension was low in the Horn of Africa. Distance takes <35 minutes was the protective factor for undiagnosed hypertension, whereas no regular exercise > = 45 years, 35–44 years, being male, good knowledge, normal BMI, overweight, poor health seeking, low vegetable consumption, smokers, high triglyceride, chat chewing, and alcohol drinking were the risk factors for undiagnosed hypertension. Therefore, health professionals and higher officials should promote a healthy lifestyle, encourage early health-seeking behaviour, and expand screening programmes.

## Supporting information

**S1 Checklist. PRISMA-2020 checklist.**
(DOCX)

**S2 Checklist. STROBE checklist.**
(DOCX)

**S1 File. A search strategy for the study of undiagnosed hypertension in the Horn of Africa.**
(DOCX)

**S2 File. Sensitivity analysis of un-diagnosed hypertension in the Horn of Africa, 2023.**
(DOCX)

**S3 File. Effect of each factor on undiagnosed hypertension in the Horn of Africa, 2023.**
(DOCX)

## Acknowledgments

The authors acknowledged University of Gondar staffs for their support.

## Author Contributions

**Conceptualization:** Muluken Chanie Agimas, Nebiyu Mekonnen Derseh, Fantu Mamo, Tilahun Yemanu.

**Data curation:** Muluken Chanie Agimas, Meron Asmamaw.

**Formal analysis:** Muluken Chanie Agimas, Nebiyu Mekonnen Derseh, Fantu Mamo, Meron Asmamaw.

**Investigation:** Muluken Chanie Agimas, Fantu Mamo, Tilahun Yemanu.

**Methodology:** Muluken Chanie Agimas, Nebiyu Mekonnen Derseh, Fantu Mamo, Moges Tadesse Abebe, Tilahun Yemanu, Meron Asmamaw.

**Resources:** Muluken Chanie Agimas.

**Software:** Muluken Chanie Agimas, Tilahun Yemanu.

**Supervision:** Tilahun Yemanu, Meron Asmamaw.

**Validation:** Moges Tadesse Abebe, Tilahun Yemanu.

**Visualization:** Meron Asmamaw.

**Writing – original draft:** Moges Tadesse Abebe, Tilahun Yemanu, Meron Asmamaw.

**Writing – review & editing:** Moges Tadesse Abebe.

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
