## [Decision Letter · Decision Letter 0]

4 Apr 2024

PONE-D-24-00667Trend, burden and determinants of undiagnosed hypertension in the horn of Africa: A systematic review and meta-analysis.PLOS ONE

Dear Dr. Agimas,

Thank you for submitting your manuscript to PLOS ONE. After careful consideration, we feel that it has merit but does not fully meet PLOS ONE’s publication criteria as it currently stands. Therefore, we invite you to submit a revised version of the manuscript that addresses the points raised during the review process.

We look forward to receiving your revised manuscript.

Kind regards,

Balew Arega Negatie, Msc,MD

Academic Editor

PLOS ONE

Journal Requirements:

2. Please include your tables as part of your main manuscript and remove the individual files. Please note that supplementary tables (should remain/ be uploaded) as separate ""supporting information"" files

Reviewers' comments:

Reviewer's Responses to Questions

**Comments to the Author**

1. Is the manuscript technically sound, and do the data support the conclusions?

Reviewer #1: Yes

Reviewer #2: No

2. Has the statistical analysis been performed appropriately and rigorously? 

Reviewer #1: Yes

Reviewer #2: No

3. Have the authors made all data underlying the findings in their manuscript fully available?

Reviewer #1: Yes

Reviewer #2: Yes

4. Is the manuscript presented in an intelligible fashion and written in standard English?

Reviewer #1: No

Reviewer #2: No

5. Review Comments to the Author

Reviewer #1: In general, your study appears to be well-conducted. However, I have a question regarding the role of regular exercise, low vegetable consumption, smoking, high TGL, chat chewing, and alcohol consumption in causing the higher prevalence of undiagnosed hypertension. Although these factors are known risk factors for the development of hypertension, they may not directly cause a person to remain undiagnosed. Therefore, it might be necessary to revise the factors that specifically contribute to the presence of undiagnosed hypertension, given that one of your research objectives is to identify the factors influencing the prevalence of undiagnosed hypertension.

Reviewer #2: After a comprehensive review of the manuscript detailing a systematic review and meta-analysis on undiagnosed hypertension in the Horn of Africa, I commend the authors for their efforts to align with the PRISMA-P 2015 guidelines and for their broad search strategy. However, several methodological and reporting concerns necessitate substantial revisions before further consideration for publication.

Search Strategy and Databases: The inclusion of Google and Google Scholar raises reproducibility issues. Prioritizing peer-reviewed databases would enhance scientific rigor.

Duplicate Records: The high volume of duplicates suggests a need for a more precise search strategy or a review of the deduplication process. Utilizing EndNote's advanced deduplication features could improve accuracy.

Study Selection and Quality Appraisal: While the approach of paired author screening is robust, a detailed account of disagreements and their resolutions would add transparency to the selection process.

Eligibility Criteria and Outcomes: The clear definition of inclusion/exclusion criteria and outcomes is a strength. Further elaboration on the identification and exclusion of studies not reporting relevant outcomes would be beneficial.

Data Extraction: The methodology for data extraction and discrepancy resolution among authors should be described in more detail to underscore the review's systematic nature.

Data Analysis: The heterogeneity assessment and the choice of modeling require a deeper discussion. Exploring the impact of study design heterogeneity on the pooled effect size is crucial.

Search Period and Typographic Errors: The mention of "February 30, 2023," along with other typographical errors, undermines the manuscript's precision and professionalism. Thorough proofreading is imperative.

Given these concerns, particularly the blending of diverse study designs without adequate stratification and the need for clearer methodological detailing, I recommend significant revisions. It is crucial that the authors address the identified issues to ensure the reliability and validity of their findings. This manuscript has the potential to contribute valuable insights into the prevalence of undiagnosed hypertension in the Horn of Africa, but as it stands, I cannot recommend it for publication without further refinement.

6. PLOS authors have the option to publish the peer review history of their article (what does this mean?). If published, this will include your full peer review and any attached files.

Reviewer #1: **Yes: **Sitotaw Kerie Bogale

Reviewer #2: **Yes: **Jorge Emilio Salazar Florez

---

## [Editor Report · Decision Letter 1]

15 Apr 2024

PONE-D-24-00667R1Trend, burden and determinants of undiagnosed hypertension in the horn of Africa: A systematic review and meta-analysis.PLOS ONE

Dear Dr. Agimas,

Thank you for submitting your manuscript to PLOS ONE. After careful consideration, we feel that it has merit but does not fully meet PLOS ONE’s publication criteria as it currently stands. Therefore, we invite you to submit a revised version of the manuscript that addresses the points raised during the review process. Please submit your revised manuscript by May 30 2024 11:59PM. If you will need more time than this to complete your revisions, please reply to this message or contact the journal office at plosone@plos.org. We look forward to receiving your revised manuscript.

Kind regards,

Balew Arega Negatie, Msc,MD

Academic Editor

PLOS ONE

**Additional Editor Comments:**

Please see the comments and suggestions in the main text

- - - - -

---

## [Editor Report · Decision Letter 2]

3 May 2024

Trend, burden and determinants of undiagnosed hypertension in the horn of Africa: A systematic review and meta-analysis.

PONE-D-24-00667R2

Dear Dr.Muluken

We’re pleased to inform you that your manuscript has been judged scientifically suitable for publication and will be formally accepted for publication once it meets all outstanding technical requirements.

Kind regards,

Balew Arega Negatie, Msc,MD

Academic Editor

PLOS ONE
---

## [Editor Report · Acceptance letter]

14 Aug 2024

PONE-D-24-00667R2 

PLOS ONE

Dear Dr. Agimas, 

I'm pleased to inform you that your manuscript has been deemed suitable for publication in PLOS ONE. Congratulations! Your manuscript is now being handed over to our production team.

Kind regards, 

on behalf of

Dr. Balew Arega Negatie 

Academic Editor

PLOS ONE